# Immunogenicity and Safety of Homologous and Heterologous Prime–Boost Immunization with COVID-19 Vaccine: Systematic Review and Meta-Analysis

**DOI:** 10.3390/vaccines10050798

**Published:** 2022-05-18

**Authors:** Haoyue Cheng, Zhicheng Peng, Shuting Si, Xialidan Alifu, Haibo Zhou, Peihan Chi, Yan Zhuang, Minjia Mo, Yunxian Yu

**Affiliations:** 1Department of Public Health and Department of Anesthesiology, The Second Affiliated Hospital of Zhejiang University School of Medicine, Hangzhou 310009, China; 3150101365@zju.edu.cn (H.C.); 22018678@zju.edu.cn (Z.P.); 21818499@zju.edu.cn (S.S.); 3130100017@zju.edu.cn (X.A.); 11918158@zju.edu.cn (H.Z.); 22118872@zju.edu.cn (P.C.); yanzhuang@zju.edu.cn (Y.Z.); minjiamo@zju.edu.cn (M.M.); 2Department of Epidemiology & Health Statistics, School of Public Health, School of Medicine, Zhejiang University, Hangzhou 310027, China

**Keywords:** COVID-19 vaccine, booster, homologous, heterologous, immunogenicity, safety

## Abstract

A prime–boost strategy of COVID-19 vaccines brings hope to limit the spread of SARS-CoV-2, while the immunogenicity of the vaccines is waning over time. Whether a booster dose of vaccine is needed has become a widely controversial issue. However, no published meta-analysis has focused on the issue. Therefore, this study assessed the immunogenicity and safety of the different combinations of prime–boost vaccinations. Electronic databases including PubMed, the Cochrane Library, Embase, medRxiv, Wanfang and CNKI were used to retrieve the original studies. A total of 28 studies, 9 combinations of prime–boost vaccinations and 5870 subjects were included in the meta-analysis, and random effect models were used to estimate pooled immunogenicity and safety. The immunity against COVID-19 after the prime vaccination waned over time, especially in the populations primed with inactivated vaccines, in which the seropositive rate of antibodies was only 28% (95% CI: 17–40%). Booster vaccination could significantly increase the antibody responses, and heterologous immunization was more effective than homologous immunization (neutralization titers: 1.65 vs. 1.27; anti-RBD IgG: 1.85 vs. 1.15); in particular, the combination of inactivated–mRNA vaccines had the highest antibody responses (neutralization titers: MRAW = 3.64, 95% CI: 3.54–3.74; anti-RBD IgG: 3.73, 95% CI: 3.59–3.87). Moreover, compared with the initial two doses of vaccines, a booster dose did not induce additional or severe adverse events. The administration of the booster dose effectively recalled specific immune responses to SARS-CoV-2 and increased antibody levels, especially in heterologous immunization. Considering the long-term immunogenicity and vaccine equity, we suggest that now, only individuals primed with inactivated vaccines require a booster dose.

## 1. Introduction

The COVID-19 pandemic has led to a dramatic loss of human life and impacted the world in terms of health, society, and economy [1]. Experts pointed out that immunity through vaccination is critical to reducing the burden of disease to relieve the pressure on governments and the subsequent economic recovery [2]. Multiple effective vaccines are being deployed globally, such as CoronaVac, BNT162b2, ChAdOx1 nCoV-19 and so on. As of 1 April 2022, there were more than ten billion vaccine doses administered worldwide (Johns Hopkins University Coronavirus Resource Center, https://coronavirus.jhu.edu/map.html accessed on 30 March 2022). Nevertheless, it cannot be ignored that serum antibody levels decrease within a few months after the completion of the prime vaccination, thereby reducing its protective effect against the COVID-19 infection [3,4,5]. A meta-analysis found that vaccine efficacy or effectiveness against SARS-CoV-2 infection decreased from one month to six months after full vaccination (two weeks after the second shot of a two-dose vaccine or after a single-dose vaccine) by 21% (95% CI: 13.9–29.8) among people of all ages [6]. Moreover, despite the high coverage of the primary vaccination in some countries, the spread of COVID-19 in these countries is still not well-controlled.

Therefore, more and more countries are recommending a booster dose of a COVID-19 vaccine to the general public [7]. Since August 2021, Israel has already promoted a third dose of the BNT162b2 vaccine to all individuals aged 12 years and older [8]. Moreover, WHO also reported that about 20% of COVID-19 vaccine doses, daily, were used for booster or additional vaccination in the world [9]. However, although the calls to offer booster doses to the public are becoming frequent, some researchers remain conservative about booster vaccinations [10]. In addition, several governments are waiting for more data before making a final decision on whether to recommend a booster dose [11].

To our knowledge, although studies on COVID-19 booster vaccinations have been carried out in many countries, no comprehensive systematic review and meta-analysis has been published focusing on the immunogenicity and safety of the booster dose. Our study will be the first one summarizing the clinical trials of the COVID-19 booster dose to compare their immunogenicity and safety, providing a useful reference for the recommendation of booster vaccinations.

## 2. Materials and Methods

### 2.1. Search Strategy and Protocol

The meta-analysis was conducted in accordance with the Cochrane Handbook for Systematic Review of Interventions [12] and reported according to the guidelines of the Preferred Reporting Items for Systematic Reviews and Meta-Analyses (PRISMA 2020 statement) [13]. The search was performed in PubMed, the Cochrane Library, Embase, medRxiv, Wanfang and CNKI to identify all published and pre-publication studies, using the key terms “COVID-19”, “vaccin*” and “booster”. Detailed search strategies for all four databases are provided in the Appendix A.

### 2.2. Eligibility Criteria

The PICOS (population, intervention, comparison, outcome and study design) approach was used to define study eligibility criteria [14]:Population—Subjects received primary COVID-19 vaccination and had no history of laboratory-confirmed COVID-19;Intervention—Booster dose of the COVID-19 vaccines;Comparison—Before and after the booster vaccination, without a control group;Outcomes—Antibody responses were assessed on the basis of the increase of antibody concentrations and the levels of antibodies at 14/28 days after booster vaccination. The secondary outcome was long-term immunogenicity after prime vaccination and booster dose safety, including adverse events at the injection site and systemic adverse events.Study designs—The articles with a before–after study design were eligible for inclusion. Animal studies, case reports, reviews, editorials and conference abstracts were excluded. Additionally, studies were excluded if there was an overlap in subjects with another study within the same analysis.

We excluded studies that did not specify the type of COVID-19 vaccines or were not published in English or Chinese. Moreover, if the interval between the first immunogenicity blood sampling (used to evaluate the baseline antibodies) and the booster vaccination was more than seven days, studies were also excluded.

### 2.3. Data Extraction and Quality Assessment

Two authors (Haoyue Cheng and Zhicheng Peng) were independently responsible for data extraction and quality assessment, and disagreements were determined by the third author (Yunxian Yu). For each study, we extracted data on study characteristics (e.g., date of publication, author names, study design, sample size, country), population demographics (e.g., sex ratio, mean age, inclusion criteria, prime–boost vaccination regimen) and outcomes (including immunogenicity of booster vaccinations and incidence of adverse events).

The quality of the included studies was independently evaluated using the Newcastle–Ottawa Scale (NOS), designed for observational and non-randomized studies [15]. The NOS contains three categories (eight subcategories), with a maximum of ten stars awardable. Scores of 0–3, 4–6 and 7–10 stars were considered a low-quality study, moderate-quality study, and high-quality study, respectively.

### 2.4. Outcomes

Outcome measures of the meta-analysis consist of three parts: antibody responses from booster vaccination, long-term immunogenicity after prime vaccination and booster dose safety. Indicators of immunogenicity included pseudotype virus neutralization titers, anti-RBD IgG concentration and the seropositive rate of antibodies. All of the indicators were measured on day 0 and day 14/28 after the booster dose. The safety outcomes were evaluated through the incidence of adverse events, including local adverse events and systemic adverse events, extracted from the original studies. Local adverse events, including injection site pain, redness, swelling and so on, occurred at the injection site. In addition, systemic adverse events were defined as those events occurring in tissues distant from the injection site, including fever, headache, body aches, fatigue and so on [16].

### 2.5. Data Synthesis and Statistical Analysis

Data analysis was conducted as recommended in the Cochrane Handbook for Systematic Reviews of Interventions [12]. Before the analysis, antibody data were log-transformed (Log10) and converted to the arithmetic mean. Due to the concentrations of anti-RBD IgG being converted to the binding antibody units/mL (BAU/mL) in the original studies, we used mean difference (MD) rather than standardized mean difference to evaluate the change in antibody responses.

Forest plots were constructed showing the summary and 95% CI estimated in the meta-analysis. The magnitude of between-study heterogeneity was estimated using the I^2^ statistical parameter. We used random effect models with inverse variance weighting, as we expected variation in effects due to differences in study populations and the methods of antibody tests. All pooled outcomes were stratified across groups of vaccination regimens. Moreover, subgroup analyses were performed to compare the differences in antibody responses across age groups. To identify a potential publication bias, Begg’s tests were conducted with different outcomes (Appendix A).

The two-independent-sample *t* test was used to compare the differences of immunogenicity between different groups. *p*  <  0.05 was considered statistically significant. All the statistical analyses were conducted using R statistical software VERSION 4.0.0 (The R Project for Statistical Computing; https://www.r-project.org).

## 3. Results

### 3.1. Characteristics of the Studies

A total of 3173 articles from PubMed (1577), the Cochrane Library (191), Embase (539), medRxiv (734), Wanfang (68) and CNKI (64) were initially included. After screening 2578 titles and abstracts and 83 full-text articles, 28 studies [17,18,19,20,21,22,23,24,25,26,27,28,29,30,31,32,33,34,35,36,37,38,39,40,41,42,43,44] provided data on nine combinations of COVID-19 booster vaccinations (including homologous immunization and heterologous immunization), and 5870 subjects met the eligibility criteria (Figure 1). The study populations mainly included the general population, health care workers and residents of a care home, which were all without history of laboratory-confirmed COVID-19. Therefore, the antibody responses were totally induced by the vaccines. All twenty-eight studies were before–after studies, and eight COVID-19 vaccines were involved: two inactivated vaccines (CoronaVac and BBIBP-CorV), two mRNA vaccines (mRNA-1273 and BNT162b2), three viral vector vaccines (ChAdOx1 nCoV-19, Ad26.COV2.S and Ad5-nCoV) and one recombinant protein vaccine (ZF2001). The nine specific groups of COVID-19 vaccination regimens were as follows (prime–boost): inactivated–inactivated, mRNA–mRNA, viral vector–viral vector, inactivated–mRNA, inactivated–viral vector, inactivated–recombinant protein, mRNA–viral vector, viral vector–inactivated and viral vector–mRNA. The main characteristics and NOS scores of all included studies are summarized in Table 1. The scores for study quality ranged from six to nine. Twenty-five studies were determined to be high-quality, three studies moderate-quality, and no study was judged as low-quality.

### 3.2. Long-Term Immunogenicity after Prime Vaccination

Before analyzing the antibody responses after the third dose of COVID-19 vaccines, the baseline level of the populations primed with different types of vaccines was also of concern. In general, all the populations still had immunity against COVID-19 at least 3 months after the prime vaccination (Figure 2, Figure 3 and Figure 4). However, compared with the published data on the immunogenicity of the vaccines, the results also confirmed that the antibody levels and clinical protective effect against COVID-19 waned over time after vaccinations without a booster dose [45,46]. Furthermore, when the vaccines were divided into three categories according to their types, the long-term immunogenicity of mRNA vaccines and viral vector vaccines was higher than that of inactivated vaccines. The baseline levels of neutralization antibody titers and anti-RBD IgG in the populations primed with mRNA vaccines were 1.93 (95% CI: 1.59–2.27) and 1.88 BAU/mL (95% CI: 1.77–2.00), respectively (Figure 2 and Figure 3). Moreover, the seropositive rate of antibodies in the populations primed with inactivated vaccines was 28% (95% CI: 17–40%), while that in the populations primed with mRNA vaccines was nearly 100% (Figure 4). In addition, the long-term immunogenicity of viral vector vaccines was similar to that of mRNA vaccines. Subgroup analyses by age found that there was no significant difference in antibody concentrations between young and old populations (Appendix A).

### 3.3. Antibody Responses after Homologous Boosters

After the homologous booster vaccines, there was a significant rise in antibody concentrations (neutralization titers: MD = 1.27, 95% CI: 1.15–1.40; anti-RBD IgG: MD = 1.15 BAU/mL, 95% CI: 0.85–1.45), and the seropositive rate of antibodies increased to almost 100% (Figure 5, Figure 6 and Figure 7). In this meta-analysis, homologous vaccination was divided into three groups: inactivated–inactivated, mRNA–mRNA and viral vector–viral vector. It is worth noting that mRNA vaccines could induce the most effective antibody responses against SARS-CoV-2 in the populations (neutralization titers: MD = 1.37, 95% CI: 1.20–1.54; anti-RBD IgG: MD = 1.49 BAU/mL, 95% CI: 1.46–1.53), while the immunogenicity of the booster dose with viral vector vaccines may not be as good as expected (neutralization titers: MD = 0.62, 95% CI: 0.38–0.86; anti-RBD IgG: MD = 0.37 BAU/mL, 95% CI: 0.22–0.52) (Figure 5 and Figure 6).

Ultimately, among the populations primed with mRNA vaccines, the levels of neutralizing antibody titers and anti-RBD IgG at 14/28 days after booster vaccination increased to 3.33 (95% CI: 3.20–3.47) and 3.39 BAU/mL (95% CI: 3.31–3.48), respectively (Appendix A). In addition, the final antibody concentrations of homologous mRNA prime–boost vaccination were significantly higher than that of the other two types of vaccines (*p* < 0.05).

### 3.4. Antibody Responses after Heterologous Boosters

A total of seven groups of heterologous vaccination regimens were included in this meta-analysis, and most populations were boosted with mRNA or viral vector after inactivated prime. From baseline to day 14/28 after the heterologous booster vaccinations, all groups had a substantial rise in antibody concentrations, and the seropositive rate of antibodies increased to almost 100% (Figure 8, Figure 9 and Figure 10). In general, heterologous immunization induced significantly higher antibody responses at 14/28 days after booster vaccination compared with homologous immunization (*p* < 0·01): the increase in anti-RBD IgG was 1.85 BAU/mL (95% CI: 1.55–2.15) for heterologous boost versus 1.15 BAU/mL (95% CI: 0.85–1.45) for homologous boost (Figure 6 and Figure 9). We also analyzed the discrepancies between different heterologous vaccination regimens. The populations boosted with heterologous boosters after inactivated vaccines had higher increases in neutralizing antibodies and anti-RBD IgG than those primed with other types of vaccines (Figure 8 and Figure 9). In addition, the results indicated that the mRNA–viral vector group induced the lowest increase in antibody levels compared to the other groups (MD = 0.93, 95% CI: 0.65–1.20).

At 14/28 days post-boost, the populations boosted with mRNA vaccines after an inactivated vaccine prime had the highest neutralization titers (MRAW = 3.64, 95% CI: 3.54–3.74) and anti-RBD IgG (MRAW = 3.73 BAU/mL, 95% CI: 3.59–3.87) (Appendix A). By contrast, boosting with inactivated vaccines may not be able to improve the antibody responses, but the number of relevant studies was too small to obtain a stable result. Moreover, the result of subgroup analyses implied that age did not affect the immunogenicity of the booster vaccines, regardless of homologous or heterologous immunization (Appendix A).

### 3.5. Booster Dose Safety

Concerning safety, we collected the data on total adverse events and divided them into local adverse events and systemic adverse events. The incidence of total adverse events (72% vs. 37%) and local adverse events (79% vs. 51%) was statistically higher in the homologous vaccination group compared with the heterologous vaccination group after the booster dose (Figure 11, Figure 12 and Figure 13). Moreover, no matter which type of vaccination regimen was applied, as long as the booster dose was mRNA or virus vector vaccines, the incidence of adverse events in the populations was higher. It may indicate that the adverse events were directly related to the mRNA and virus vector vaccines, but not to the prime–boost vaccination regimens.

## 4. Discussion

In this meta-analysis of before–after studies, we found that the immunity against COVID-19 after the prime vaccination waned over time, especially the long-term immunogenicity of the inactivated vaccines. Booster vaccination could significantly ameliorate the antibody responses, and heterologous immunization was more effective than homologous immunization. A heterologous prime–boost regimen with inactivated vaccine followed by an mRNA vaccine boost markedly increased the antibody concentrations, which may be the most effective vaccination strategy. Moreover, compared with the initial two doses of vaccines, a booster dose did not induce additional or severe adverse events.

Lowering the rates of infection help break the cycle of viral transmission, which can eventually reduce cases of severe COVID-19 and death [47]. However, up to now, multiple studies have indicated a decrease in the immunogenicity of COVID-19 vaccines over time [4,5,48]. A meta-analysis evaluating four vaccines further found that, although the vaccine efficacy against severe disease remained high (≥70%) for up to 6 months after vaccination, the decline of vaccine efficacy against SARS-CoV-2 infection could not be ignored [6]. Waning antibody concentrations is a plausible explanation for the decrease in vaccine efficacy against infection and disease [47]. The results of this meta-analysis are consistent with this explanation, especially the alarming decline in the seropositive rate of antibodies of the inactivated vaccines. Moreover, Feikin et al. [6] indicated that the decrease in vaccine efficacy over time was not caused by the SARS-CoV-2 variants. To sum up, from the perspective of maintaining the immunogenicity of the COVID-19 vaccines, a booster dose is critical.

Humoral immunity and cell-mediated immunity are two types of adaptive immune responses that enable the human body to defend itself against SARS-CoV-2. However, neutralizing antibodies that can intercept viruses before they penetrate cells do not have much staying power, while cellular immune responses are longer-lasting [47]. Therefore, a booster dose is a “trigger” that can stimulate B cells to secrete more neutralizing antibodies, so as to prevent the invasion of SARS-CoV-2 [49]. The results of this meta-analysis confirmed the benefit of booster vaccinations, and it also indicated that the immunogenicity of heterologous immunization was much better. The mechanism for this difference is that using dissimilar platforms can induce protection from different pathways [50]. Different types of vaccines have their own advantages. The theoretical advantage of inactivated vaccines is that they contain additional viral proteins, such as nucleoprotein, which can potentially extend the protection beyond anti-spike protein responses [26]. The mRNA vaccines are able to elicit extremely high neutralizing and binding antibody titers (especially the anti-spike IgG), while the vector-based vaccines produce polyclonal antibodies [51,52]. Palgen et al. [53] indicated another plausible mechanism that could explain the better immunogenicity of heterologous boosters. Preexisting trained innate cells and antibodies to the same vaccine tend to impair antigen presentation in individuals boosted with homologous vaccines. However, when an unrelated heterologous vaccine is administered, cells may produce subsequent robust responses of naive cells via epigenetic reprogramming. Unfortunately, since the original studies did not provide enough data on B cell and T cell responses, the mechanism could not be verified in this meta-analysis. In addition to the type of COVID-19 booster vaccines, Chiu et al. [54] found that the order of prime–boost also mattered. In addition, the results of our study showed that the immunogenicity of the viral vector–mRNA vaccination regimen was better than that of the mRNA–viral vector vaccination regimen. Therefore, further studies are required to clarify the underlying mechanisms and the best order of vaccinations.

The meta-analysis summarized the immunogenicity and safety of the COVID-19 booster vaccines in healthy populations without a history of laboratory-confirmed COVID-19. However, it has several limitations. First, most of the original studies were limited in several countries where vaccination was widely promoted, such as China, USA and Thailand. In addition, due to government policies, individuals in a country always accepted a certain type of vaccine. Therefore, the representativeness of the meta-analysis results may be affected by race, vaccination strategy and so on. Second, the interval between prime and boost may influence the efficacy of COVID-19 vaccines [54,55]. However, most original studies did not provide information about the interval, which induced the lack of the relevant subgroup analyses. Third, when the studies were stratified by the type of vaccines, the number of studies in each group was small. Therefore, achieving adequate statistical power may be difficult, and a cautious approach in interpreting the results is warranted. Fourth, although there was some consistency between different measurement methods, specific processes and laboratory equipment varied in different studies. It may be less accurate to directly compare the immunogenicity of different studies [56].

In general, our study is the first meta-analysis confirming the immunogenicity and safety of COVID-19 booster vaccination, especially the superiority of heterologous immunization. However, any discussion around the requirement of boosters cannot be had in a vacuum. Vaccine equity remains an issue that cannot be ignored. At the current rate of vaccination, low-income countries are unable to achieve substantial protection levels until at least 2023 [57]. This situation is not conducive to controlling the worldwide spread of COVID-19 and will drive SARS-CoV-2 evolution [58]. Therefore, it is critical to find a balance between vaccine equity and booster vaccination.

## 5. Conclusions

There is no doubt that the administration of the booster dose effectively recalls specific immune responses to SARS-CoV-2 and increases antibody levels, and heterologous immunization is more effective than homologous immunization. Considering long-term immunogenicity and vaccine equity, we suggest that a booster dose be required in individuals primed with inactivated vaccines, while individuals primed with other types of vaccines can appropriately hold off on the administration of boosters.

## Figures and Tables

**Figure 1 vaccines-10-00798-f001:**
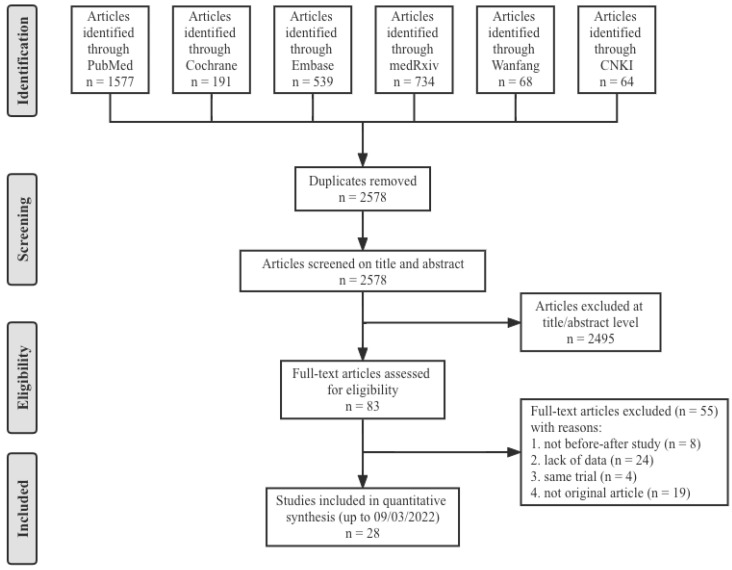
Flow diagram showing the progress through the stages of meta-analysis.

**Figure 2 vaccines-10-00798-f002:**
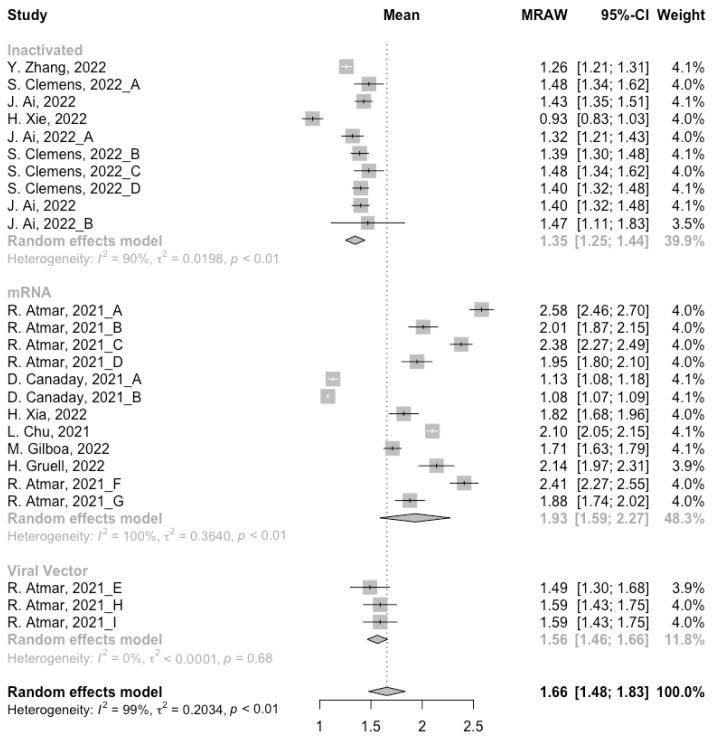
Forest plot of the pooled log-transformed neutralization antibody titers before booster vaccination.

**Figure 3 vaccines-10-00798-f003:**
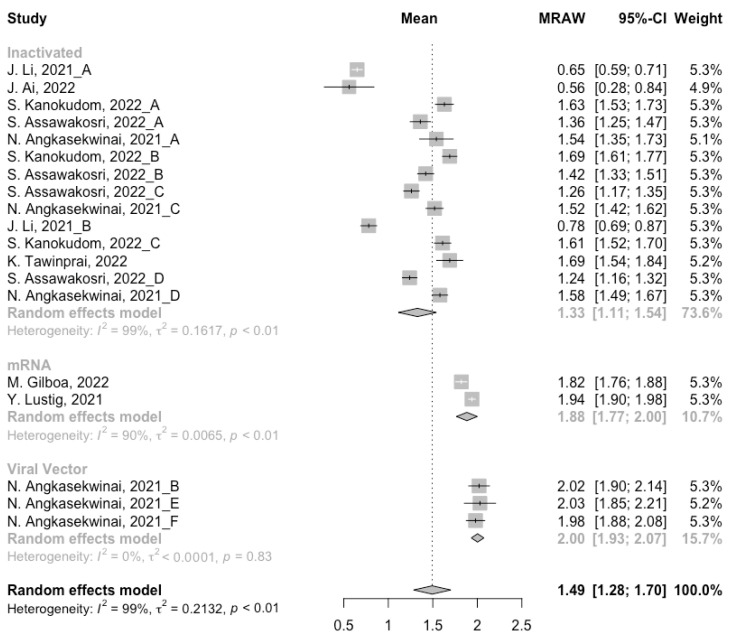
Forest plot of the pooled log-transformed anti-RBD IgG before booster vaccination.

**Figure 4 vaccines-10-00798-f004:**
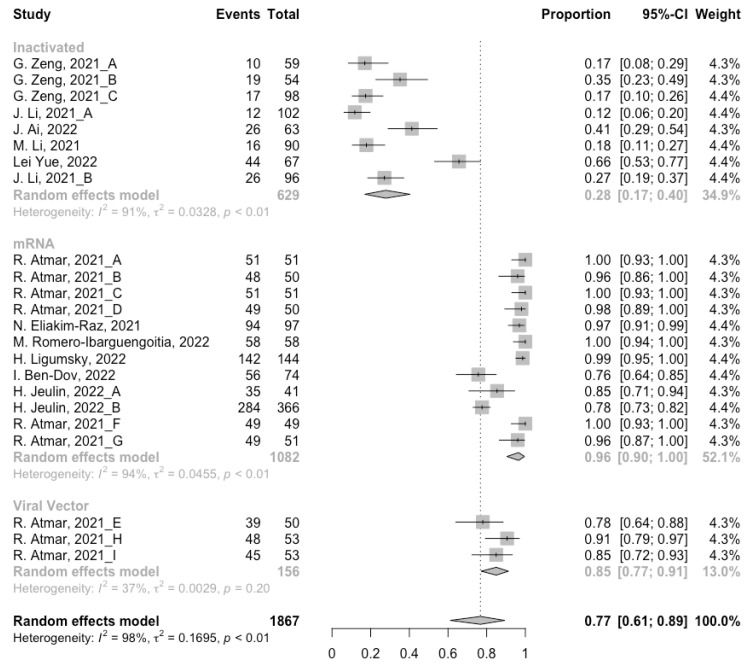
Forest plot of the pooled seropositive rate of antibodies before booster vaccination.

**Figure 5 vaccines-10-00798-f005:**
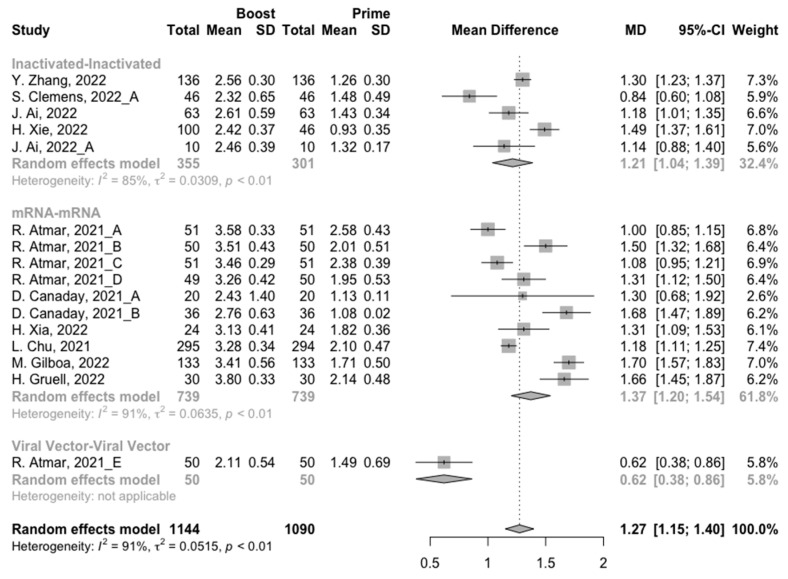
Forest plot of the pooled log-transformed neutralization antibody titers before and after homologous booster vaccination.

**Figure 6 vaccines-10-00798-f006:**
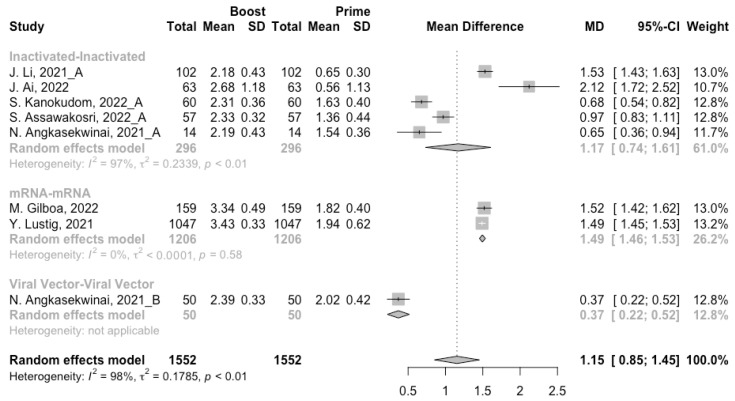
Forest plot of the pooled log-transformed anti-RBD IgG before and after homologous booster vaccination.

**Figure 7 vaccines-10-00798-f007:**
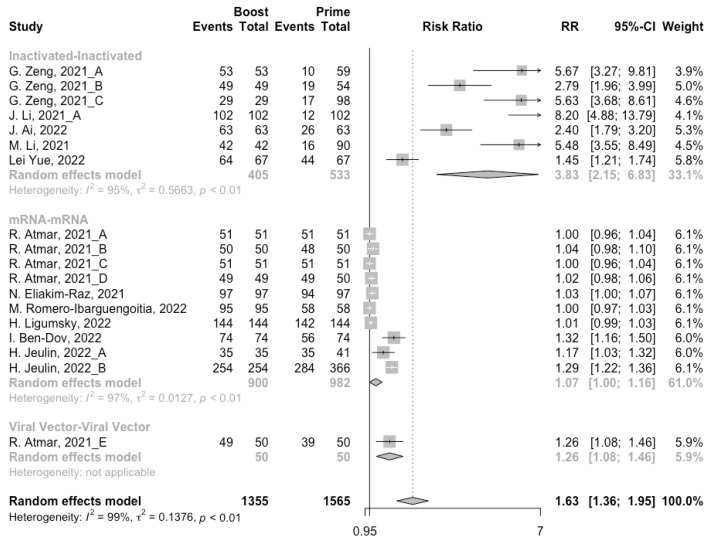
Forest plot of the pooled seropositive rate of antibodies before and after homologous booster vaccination.

**Figure 8 vaccines-10-00798-f008:**
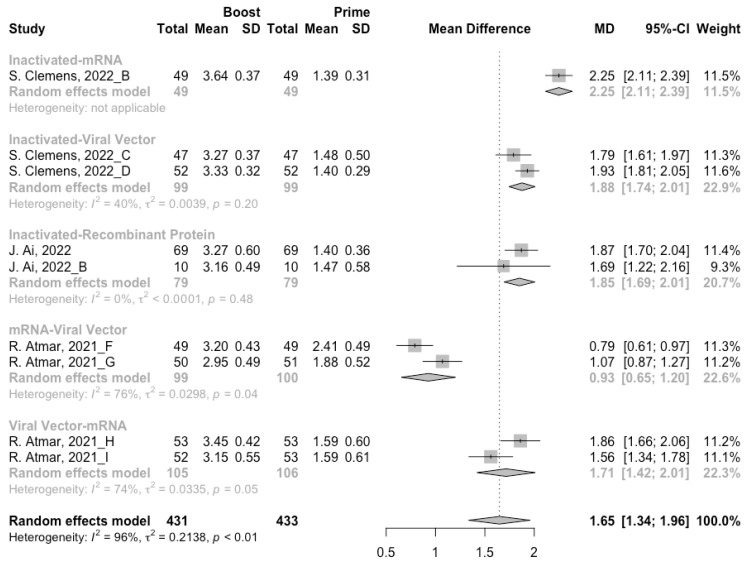
Forest plot of the pooled log-transformed neutralization antibody titers before and after heterologous booster vaccination.

**Figure 9 vaccines-10-00798-f009:**
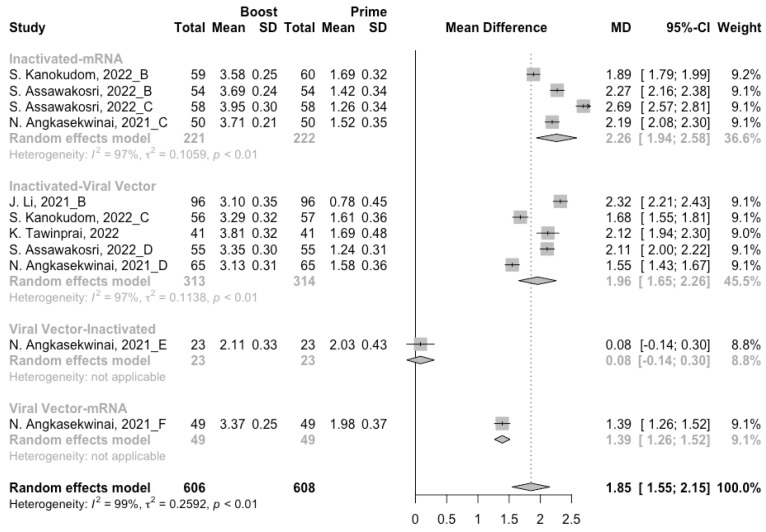
Forest plot of the pooled log-transformed anti-RBD IgG before and after heterologous booster vaccination.

**Figure 10 vaccines-10-00798-f010:**
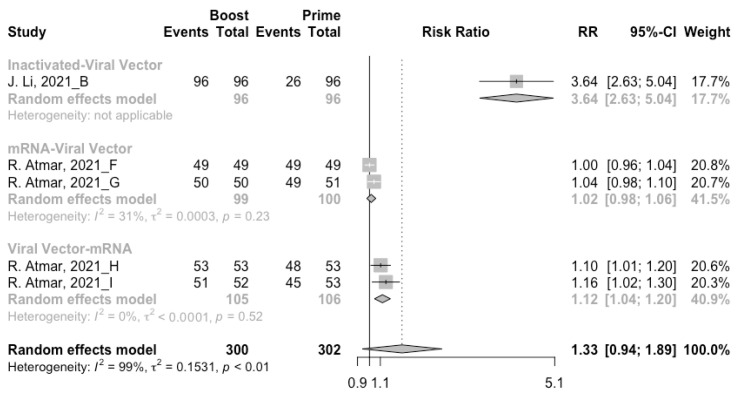
Forest plot of the pooled seropositive rate of antibodies before and after heterologous booster vaccination.

**Figure 11 vaccines-10-00798-f011:**
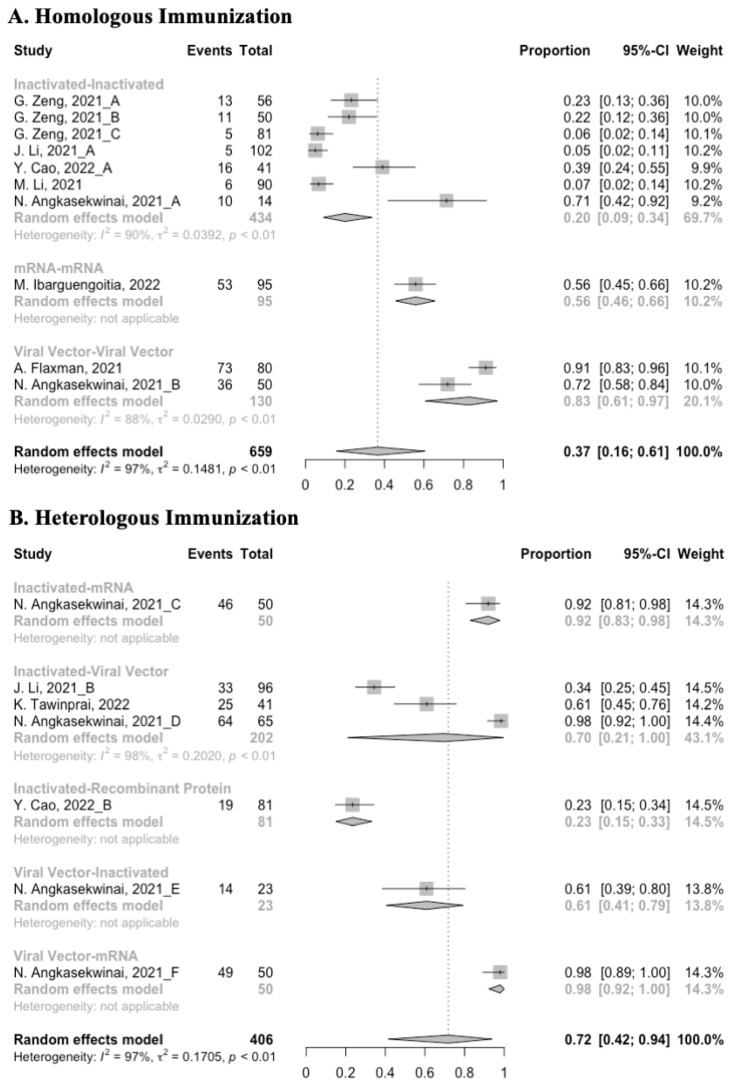
Forest plot of the overall incidence of total adverse events after booster vaccination.

**Figure 12 vaccines-10-00798-f012:**
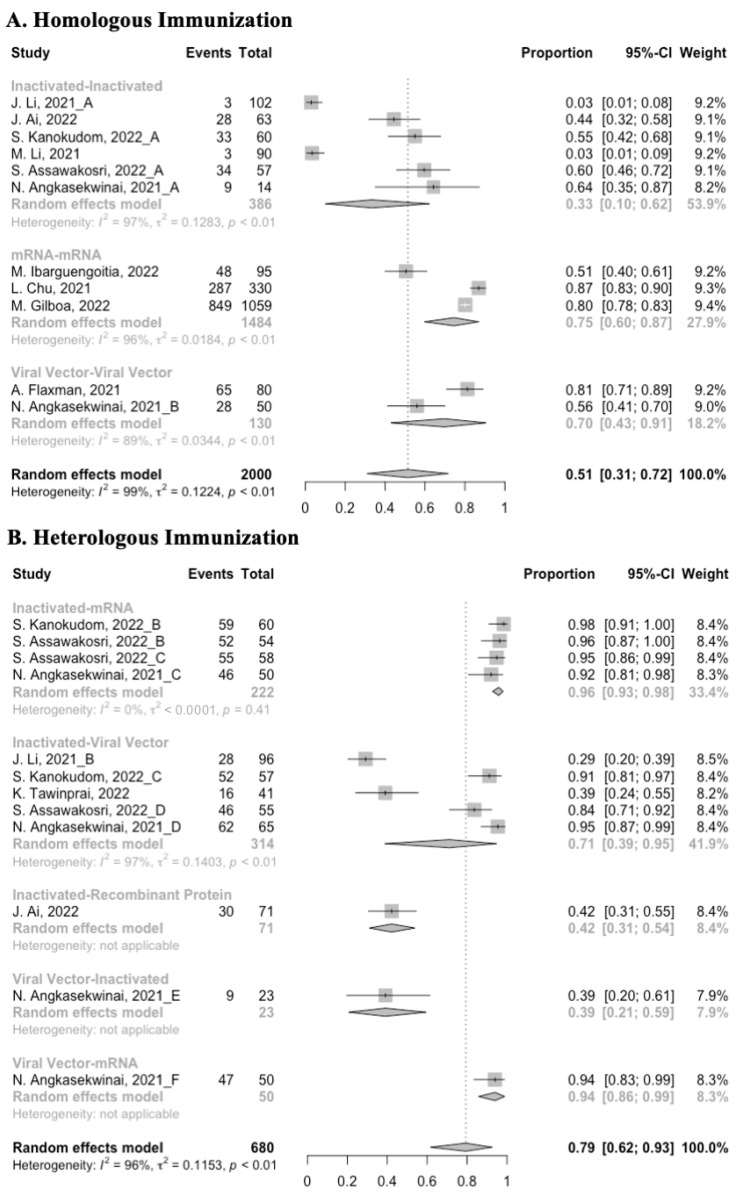
Forest plot of the overall incidence of local adverse events after booster vaccination.

**Figure 13 vaccines-10-00798-f013:**
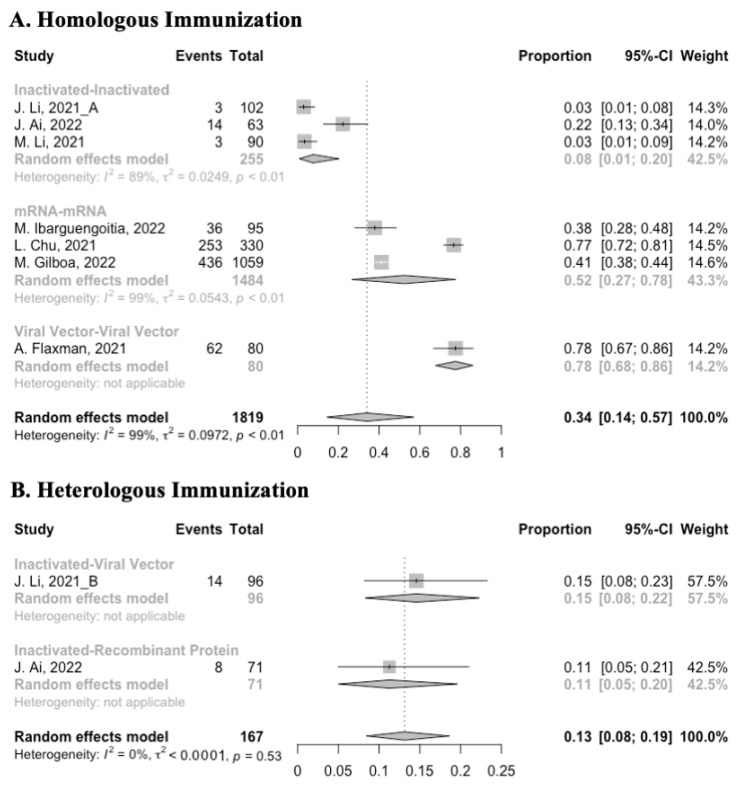
Forest plot of the overall incidence of systemic adverse events after booster vaccination.

**Table 1 vaccines-10-00798-t001:** Characteristics of the original studies included in the meta-analysis.

Study and Year	Country	Number of Groups	Participants (N)	Characteristics of the Participants 1	Age (Mean/Median)	Male (%)	COVID-19 Vaccines (Prime/Boost) 2	Interval of Boost	Antibody Detection Method	NOS Score
Zeng et al., 2021	China	3	59; 54; 98	Healthy adults aged 18–59 years or 60 years and older	40.4; 44.3; 66.4	44; 44; 49	CoronaVac/ CoronaVac	8 months	Seropositive rate: micro cytopathic effect assay	7
Atmar et al., 2021	USA	9	51; 50; 51; 50; 50; 49; 51; 53; 53	Healthy adults	53.1; 54.8; 54.3; 50.4; 50.1; 49.9; 50.3; 56.8; 47.7	37.3; 42; 49; 54; 54; 67.3; 54.9; 50.9; 45.3	mRNA-1273/mRNA-1273; BNT/mRNA-1273; mRNA-1273/BNT; BNT/BNT; Ad26/Ad26; mRNA-1273/Ad26; BNT/Ad26; Ad26/mRNA-1273; Ad26/BNT	at least 12 weeks	Neutralization titers: pseudovirus	8
Li et al., 2021	China	2	102; 96	Healthy adults aged 18–59 years	45.4; 44.8	62.8; 60.4	CoronaVac/ CoronaVac; CoronaVac/Ad5	3–6 months	Seropositive rate: micro cytopathic effect assay; anti-RBD IgG: ELISA	9
Flaxman et al., 2021	UK	1	75	Healthy adults	37	60	ChAd/ChAd	20–38 weeks	-	6
Canaday et al., 2021	USA	2	29; 53	Healthy nursing home residents or health care workers	50; 75	59; 70	BNT/BNT	6–8 months	Neutralization titers: pseudovirus	7
Cao et al., 2022	China	2	41; 81	Healthy adults	38.1; 40.7	24.4; 30.9	CoronaVac/ CoronaVac; CoronaVac/ZF2001	4–8 months	-	9
Eliakim-Raz et al., 2021	Israel	1	97	Healthy adults aged 60 years and older, without active malignancy	70	39	BNT/BNT	NA	Seropositive rate: chemiluminescent microparticle immunoassay	7
Ai et al., 2022	China	1	69	Healthy adults	28	43.7	BBIBP/ZF2001	4–8 months	Neutralization titers: pseudovirus	9
Zhang et al., 2022	China	1	136	Healthy adults	38	52.9	BBIBP/BBIBP	6–14 months	Neutralization titers: pseudovirus	8
Clemens et al., 2022	Brazil	4	281; 333; 295; 296	Healthy adults	60	39.5	CoronaVac/ CoronaVac; CoronaVac/BNT; CoronaVac/Ad26; CoronaVac/ChAd	6 months	Neutralization titers: pseudovirus	9
Ai et al., 2022	China	1	63	Healthy adults	28	42.9	BBIBP/BBIBP	4–8 months	Neutralization titers: pseudovirus; anti-RBD IgG: chemiluminescent immunoassay	9
Xie et al., 2022	China	1	46	Healthy adults aged 18–59 years	NA	NA	CoronaVac/ CoronaVac	at least 12 months	Neutralization titers: pseudovirus	8
Kanokudom et al., 2022	Thailand	3	60; 60; 57	Healthy adults	42.7; 44.2; 41.6	50; 40; 50.9	CoronaVac/BBIBP; CoronaVac/BNT; CoronaVac/ChAd	3–4 months	Anti-RBD IgG: chemiluminescent microparticle immunoassay	9
Xia et al., 2022	USA	1	24	Healthy adults	52.9	37.5	BNT/BNT	NA	Neutralization titers: pseudovirus	8
Li et al., 2021	China	1	90	Healthy adults aged 60 years and older	66.4	49	CoronaVac/ CoronaVac	6 months	Seropositive rate: micro cytopathic effect assay	7
Romero-Ibarguengoitia et al., 2022	Mexico	1	58	Healthy adults	41.7	36.8	BNT/BNT	166.3 ± 12.3 days	Seropositive rate: chemiluminescent immunoassay	6
Chu et al., 2021	USA	1	295	Healthy adults	52	33.7	mRNA-1273/mRNA-1273	7.2 ± 0.6 months	Neutralization titers: pseudovirus	9
Gilboa et al., 2022	Israel	1	159	Healthy adults aged 60 years and older	66	35	BNT/BNT	NA	Neutralization titers: pseudovirus; anti-RBD IgG: chemiluminescent microparticle immunoassay	8
Yue et al., 2022	China	1	67	Healthy adults	NA	NA	inactivated/inactivated	8 months	NA	7
Tawinprai et al., 2022	Thailand	1	41	Healthy adults	45	61	CoronaVac/ChAd	at least 2 months	Anti-RBD IgG: electrochemiluminescence immunoassay	9
Gruell et al., 2022	Germany	1	30	Healthy adults	49	43	BNT/BNT	26–41 weeks	Neutralization titers: pseudovirus	7
Ligumsky et al., 2022	Israel	1	144	Healthy adults	62	34.8	BNT/BNT	at least 5 months	Seropositive rate: chemiluminescent immunoassay	8
Ben-Dov et al., 2022	Israel	1	74	Healthy adults	NA	NA	BNT/BNT	6 months	Seropositive rate: chemiluminescent immunoassay	6
Ai et al., 2022	China	2	10; 10	Healthy adults	27; 24.5	60; 60	BBIBP/BBIBP; BBIBP/ZF2001	4–8 months	Neutralization titers: pseudovirus	8
Lustig et al., 2021	Israel	1	1047	Healthy health care workers	47.7	27.1	BNT/BNT	at least 3 months	Anti-RBD IgG: chemiluminescent immunoassay	8
Jeulin et al., 2022	France	2	41; 366	Healthy adults aged 65 years and older	84; 88	37; 22	BNT/BNT	7 months	NA	7
Assawakosri et al., 2022	Thailand	4	57; 54; 58; 55	Healthy adults	41.9; 41.6; 37; 44.1	40.4; 59.3; 47.8; 43.6	CoronaVac/BBIBP; CoronaVac/BNT; CoronaVac/mRNA-1273; CoronaVac/ChAd	5–7 months	Anti-RBD IgG: electrochemiluminescence immunoassay	9
Angkasekwinai et al., 2021	Thailand	6	14; 50; 50; 65; 23; 49	Healthy adults	31; 45.5; 32; 36.6; 51; 34	14.3; 6; 20; 21.5; 8.7; 26	CoronaVac/BBIBP; ChAd/ChAd; CoronaVac/BNT; CoronaVac/ChAd; ChAd/BBIBP; ChAd/BNT	8–12 weeks	Anti-RBD IgG: chemiluminescent microparticle assay	8

^1^ All studies recruited participants without a history of laboratory-confirmed COVID-19. ^2^ BNT: BNT162b2; Ad26: Ad26.COV2.S; Ad5: Ad5-nCoV; ChAd: ChAdOx1 nCoV-19; BBIBP: BBIBP-CorV.

## Data Availability

The data presented in this study are available in the article.

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
