# Peer review of "Immunogenicity and Safety of Homologous and Heterologous Prime–Boost Immunization with COVID-19 Vaccine: Systematic Review and Meta-Analysis"

_vaccines, 2022, doi:10.3390/vaccines10050798_

Round 1

Reviewer 1 Report

The authors investigated the immunogenicity and safety of covid-19 vaccines in a systematic review and meta-analysis.

some comments should improve the manuscript:

1/ Key search terms included delivery concepts should be added in the article.

2/ Risk of bias assessment is needed:

During the data extraction process, researchers independently assessed the risk of bias for each study using the Cochrane Collaboration’s risk of bias tool. Evaluation criteria included the following: random sequence generation, allocation concealment, blinding of students and personnel, blinding of outcome assessment, incomplete outcome data, selective reporting, or other which included publication bias. Funnel plots were used to evaluate publication bias. Risk of bias for each criterion was rate as low, high, or unclear according to the Cochrane risk of bias instructions.

Ref.Higgins JPT, Altman DG, Gøtzsche PC, Jüni P, Moher D, Oxman AD, Savovic J, Schulz KF, Weeks L, Sterne JAC, Cochrane BMG, Cochrane SMG. The Cochrane Collaboration's tool for assessing risk of bias in randomised trials. BMJ. 2011;343:d5928. http://europepmc.org/abstract/MED/22008217. [PMCID: PMC3196245][PubMed: 22008217]

3/ statistically significant if P<0.05?

4/To explore publication bias, Begg tests should be performed.

5/Figure 2: for viral vector, p=0.68 but MRAW=1.56 [1.46 -1.66], are you sur for heterogeneity ? I2 and t2? =0 

figure 2: for mRNA, p=0?

figure 2: random effects model, p=0?

6/ same observation for figure 3 and viral vector (p value, I2, T2) and random effects model p=0?

7/ same observation for figure 6 and mRNA (p value, I2, T2) and random effects model p=0?

8/ same observation for figure 7 and random effects model p=0?

9/ figure 8, same observations for inactivated-viral vector/recombinant protein.

10/ same for figure 10, viral vector-mRNA p=0.52 and RR=1.12 [1.04-1.20], i2=0? t2=0?

11/ reference could be added in the article: 

  • DOI: 10.3390/jcm10173817

Author Response

Point 1: Key search terms included delivery concepts should be added in the article.

Response 1: Thank you for your suggestion. We have added the key search terms “COVID-19”, “vaccin*” and ”booster” in the manuscript.

Point 2: Risk of bias assessment is needed:

During the data extraction process, researchers independently assessed the risk of bias for each study using the Cochrane Collaboration’s risk of bias tool. Evaluation criteria included the following: random sequence generation, allocation concealment, blinding of students and personnel, blinding of outcome assessment, incomplete outcome data, selective reporting, or other which included publication bias. Funnel plots were used to evaluate publication bias. Risk of bias for each criterion was rate as low, high, or unclear according to the Cochrane risk of bias instructions.

Ref.Higgins JPT, Altman DG, Gøtzsche PC, Jüni P, Moher D, Oxman AD, Savovic J, Schulz KF, Weeks L, Sterne JAC, Cochrane BMG, Cochrane SMG. The Cochrane Collaboration's tool for assessing risk of bias in randomised trials. BMJ. 2011;343:d5928. http://europepmc.org/abstract/MED/22008217. [PMCID: PMC3196245][PubMed: 22008217]

Response 2: Sorry for making you confused. Because the included articles were all before-after study designs, we used the New-castle–Ottawa Scale (NOS) to assess the quality. And the NOS scores were summarized in Table 1.

Point 3: statistically significant if P<0.05?

Response 3: You are right. We have added the threshold of significance (P < 0.05) in the Data synthesis and statistical analysis section.

Point 4: To explore publication bias, Begg tests should be performed.

Response 4: Begg’s tests have been conducted with different outcomes (Table S5). Overall, there is no publication bias in all results except the seropositive rate of antibodies.

Point 5: Figure 2: for viral vector, p=0.68 but MRAW=1.56 [1.46 -1.66], are you sur for heterogeneity ? I2 and t2? =0; figure 2: for mRNA, p=0? figure 2: random effects model, p=0?

Point 6: same observation for figure 3 and viral vector (p value, I2, T2) and random effects model p=0?

Point 7: same observation for figure 6 and mRNA (p value, I2, T2) and random effects model p=0?

Point 8: same observation for figure 7 and random effects model p=0?

Point 9: figure 8, same observations for inactivated-viral vector/recombinant protein.

Point 10: same for figure 10, viral vector-mRNA p=0.52 and RR=1.12 [1.04-1.20], i2=0? t2=0?

Response 5-10: Thank you for your reminder. The reason for these problems is the values (P, τ2) are very close to 0, therefore R software defaulted these values to 0 when plotting. We have revised all figures as follows: P = 0 changed to P < 0.01, and τ2 = 0 changed to τ2 < 0.0001.

Point 11: reference could be added in the article: DOI: 10.3390/jcm10173817

Response 11: We have added the reference in the Discussion section.

Reviewer 2 Report

The manuscript deals with interesting topic that is attracting the attention of the public and scientists on the effectiveness of Covid-19 vaccines, especially when multiple effective vaccines are introduced and different vaccination schedules are used.

The quality of the manuscript does not rise any objections: methods are appropriate, all parts of the manuscript (methods, results and discussion) are decent and well-described. The strength of the study is deep and complex statistic analysis suitable for such kind of epidemiological research with very detailed description of partial analyses.

Author Response

Point 1: The manuscript deals with interesting topic that is attracting the attention of the public and scientists on the effectiveness of Covid-19 vaccines, especially when multiple effective vaccines are introduced and different vaccination schedules are used.

The quality of the manuscript does not rise any objections: methods are appropriate, all parts of the manuscript (methods, results and discussion) are decent and well-described. The strength of the study is deep and complex statistic analysis suitable for such kind of epidemiological research with very detailed description of partial analyses.

Response 1: Thank you for your appreciation.

Reviewer 3 Report

The manuscript by Cheng and colleagues provides a meta-analysis of 33 studies and nine prime-boost combinations of prime-boost immunization with COVID-19 vaccines.  The authors evaluated three basic types of vaccines (inactivated, mRNA and viral vector vaccines). Overall, I find that the search strategies, eligibility criteria, outcomes (antibody responses of the booster vaccination, long-term immunogenicity, and booster safety) to be acceptable. The data are presented in a series of Tables which are easy to follow and support the major conclusions of the manuscript that are: 1) that the mRNA and viral vectors vaccines provide the best primary antibody responses to booster vaccinations; 2) booster responses with homologous vaccines (i.e., mRNA-mRNA or viral vector-viral vector) are not as robust as those with heterologous prime-boost combinations; and 3) those individuals whose primary immunization was with the inactivated vaccine should obtain a booster vaccination. These results are important to vaccination protocols in the future.

Major comments:

1) The authors examine virus neutralization and antibody titers against the receptor binding domain (RBD) following the primary immunization and booster immunizations. I believe that the authors need to provide details of these neutralization assays and how the titers were determined. Importantly, were these assays performed with standard protocols or did they vary from study to study?

2) The authors should define what constitutes local and systematic adverse effects.

Minor comments:

1.Line 29: sever should be severe.

2.line 47: Please define the meaning of “full vaccination.”

3.line 215: “higher increasement of neutralising anti-” should be changed to “higher increase in neutralising anti-

4.line 117: Please define BAU.

5. Line 262: “mete” should be changed to “meta.”

6. Line 274-275: “penetrate cells do always not have  . . . ” should be changed to “penetrate cells do not have....” 

7) line 301-302: “China, USA, Thailand and so on.” should be changed to “China, USA, and Thailand.” 

Author Response

Point 1: The authors examine virus neutralization and antibody titers against the receptor binding domain (RBD) following the primary immunization and booster immunizations. I believe that the authors need to provide details of these neutralization assays and how the titers were determined. Importantly, were these assays performed with standard protocols or did they vary from study to study? 

Response 1: We have extracted the methods of antibody detection in the included studies and summarized them in Table 1. Overall, all studies used pseudovirus neutralisation assays to evaluate neutralisation titres, and most studies used chemiluminescent immunoassay to evaluate anti-RBD IgG antibody titres. However, most studies did not provide detailed protocols, so we added the issue to the limitations in the Discussion section.

Point 2: The authors should define what constitutes local and systematic adverse effects.

Response 2: Thank you for your advice. The definitions and types of local and systematic adverse events have been added as follows:

“Local adverse events, including injection site pain, redness, swelling and so on, occurred at the injection site. And systemic adverse events were defined as those events occurring in tissues distant from the injection site, including fever, headache, body aches, fatigue and so on.”

Point 3: Line 29: sever should be severe.

Response 3: Thank you for the correction. We have corrected the mistake.

Point 4: line 47: Please define the meaning of “full vaccination.”

Response 4: We have added the definition of “full vaccination” in the manuscript, which means two weeks after the second shot of a two-dose vaccine or after a single-dose vaccine.

Point 5: line 215: “higher increasement of neutralising anti-” should be changed to “higher increase in neutralising anti-

Response 5: We have corrected it based on your suggestion.

Point 6: Line 117: Please define BAU.

Response 6: We have added the full form of BAU (binding antibody units) in the manuscript.

Point 7: Line 262: “mete” should be changed to “meta.”

Response 7: Thank you for the correction. We have corrected it.

Point 8: Line 274-275: “penetrate cells do always not have  . . . ” should be changed to “penetrate cells do not have....”

Response 8: Thank you for the correction. We have modified it.

Point 9: Line 301-302: “China, USA, Thailand and so on.” should be changed to “China, USA, and Thailand.”

Response 9: Thanks. We have modified it.

Round 2

Reviewer 1 Report

Acceptance for publication.